# Reinforcement Learning for Predict+Optimize

**Xinyi HU**
Department of Computer Science and Engineering
The Chinese University of Hong Kong
Shatin, Hong Kong
xyhu@cse.cuhk.edu.hk

**Yuansen CHENG**
Department of Information Engineering
The Chinese University of Hong Kong
Shatin, Hong Kong
cy019@ie.cuhk.edu.hk

## Abstract

Predict+Optimize (P+O) is a machine learning framework for optimization problems with unknown parameters. This paper presents a framework to tackle P+O problems using neural networks and reinforcement learning. We focus on the traveling salesman problem and train a recurrent neural network that, given a directed graph, predicts a distribution over different edges permutations. Using negative tour length as the reward signal, we optimize the parameters of the recurrent neural network using a policy gradient method. The short video link is here.

## 1 Introduction

*Combinatorial Optimization Problems* (COP) is a crucial and well-researched area in computer science. However, in traditional optimization, it is assumed that all the parameters in COPs are given, which is often unrealistic. The parameters in COPs are usually hidden and are crude estimates based on domain expertise or historical data, and this kind of problem is called *Predict+Optimize problem* (P+O) (11). When dealing with this kind of problem, classical approaches usually contains two steps: first machine learning models use the historical data to predict the hidden parameters; then COPs use these predictions to arrive at a decision that maximizes some objectives. The modern techniques used in the first step may include traditional regression (6; 19), neural networks (24; 26; 1), decision trees (27), clustering (9; 5; 22), and so on.

However, some researchers (16; 18) found that the best forecast may have the worst result when employed in the COPs. The reason is that the aim of P+O is to learn a predictive model that results in a decision that maximizes the true parameters, rather than minimizes the prediction error between estimated parameters and true parameters. To make better prediction, Elmachtoub et al. (13) propose the *Smart "Predict, then Optimize"* (SPO) framework, including a novel evaluation criterion called *regret* which take COP into account. The core idea is based on supervised learning, where a mapping from training inputs to outputs is learned.

One problem that supervised learning will meet is that the regret function is non-differentiable and thus cannot be applied to gradient-based learning directly. From reinforcement learning (RL) we know, one can compare the quality of a set of solutions using a verifier, and provide some reward feedbacks to a learning algorithm. Hence, we follow the RL paradigm to tackle P+O problems. In this paper, we shall investigate the power of reinforcement learning in solving P+O problems. As far as we know, we are the first one to apply RL to P+O problems.

## 2 Related Work

Existing works on P+O area usually follow the SPO framework, and one challenge is that the regret function is non-differentiable. One straightforward method is to transform the regret function into a differentiable surrogate function via certain relaxations or conditions.

Motivating by duality theory, a scaling approximation, and a first-order approximation, Elmachtoub et al. (13) derive a surrogate function, called SPO+ loss function, from three upper bounds on SPO loss function. Balghiti et al. extend their work by providing two kinds of generalization bounds (12). Wilder et al. (30) propose another framework called *decision-focused learning*. Their aim is directly maximizing the objective values of the maximize optimization problem, instead of minimizing the loss function. They relax the COP to a continuous one and obtain the gradients of the objective with the help of the Karush-Kuhn-Tucker conditions. J. Mandy and T. Guns (23) focus on optimization problems that can be modeled as MILP/ILP/ LP. They propose to differentiate the homogeneous self-dual formulation, instead of the KKT condition and show its effectiveness. Researchers also develop some novel network structures. Ferber et al. (15) propose a novel machine learning component called MIPaaL, which extends the decision-focused learning framework to the problems that can be encoded as MIP. They formulate the MIP as an extra layer in the neural network. The input of this layer is the estimated parameters, while the output is the optimal MIP solution. Vlastelica et al. introduce a combinatorial solver into neural networks, to achieve fusion of deep learning with combinatorial algorithms (29). Unlike MIPaaL and other approaches, the combinatorial solver served as a blackbox in the networks. The input of this blackbox combinatorial solver is the learned representation (i.e. the feature vector), and the output is the optimal solution. They construct a continuous interpolation of the linearization of the loss function, to do the gradient-based backward pass.

However, the surrogate regret functions are not guaranteed to recover optimal decisions with respect to the original regret and merely serve as approximations. Therefore, some researchers try to make use of some special properties of the COPs or the ML models, so that the models can be trained under the new loss without computing gradients (10). Elmachtoub, Liang, and McNellis propose a method called SPO Trees (SPOTs) that can train decision trees using the original SPO loss function (14). Similar to Classification and Regression Trees algorithm, SPOTs use recursive partitioning to train the trees. They propose a greedy split selection approach to do the selection and splitting. Demirovic et al. (10) come up with a special property called ranking property. They point out that the solution space of these ranking COPs can be divided into a finite number of blocks according to the objective values. Solutions with the same objective value are divided into the same solution block. Then find the solution block which the optimal solution in. The key is to find the transition points or transition lines. But in this paper they do not propose an efficient method to find out the transition points. Later, they present an algorithm to support problems solvable by dynamic programming (11), in which the COP with unknown parameters is represented as a piecewise linear function, parameterized by the learning model parameters $\alpha$. Each segment of the piecewise function corresponds to a block of solution space. With the help of dynamic programming, they can easily get the transition points of each segment (or solution block).

On the other hand, recently, many researchers apply RL to solve COPs. Vinyals proposed a network structure called pointer network (PN) to solve some classic COPs, such as traveling salesman problem (TSP) and knapsack (28), which forms the basis for many papers that apply machine learning methods to COPs. They used an attention mechanism to calculate Softmax probability values, used them as pointers to elements in the input sequences, combined the input sequences, and finally trained the model using supervised methods. Bello et al. (4) use Actor-critc algorithm to train PN and obtain approximate optimal solutions on the TSP problem with node length $n = 100$. Nazari et al. (25) apply RL paradigm to tackle vehicle routing problem. They replace PN's encoder part with an embedding layer, so that when dynamic elements in the input sequence change, it is not necessary to completely update the RNN part of the encoder and decrease the computation. Based on Self Attention (MHA+FF) in Transformer, W. Kool et al. (21) establish a complex Encoder and USES Rollout enhanced learning algorithm to train the model and solve TSP, VRP and their variants OP, PCTSP and SPCTSP. Besides, deep Q-learning network is also one of the most popular methods for solving many COPs. Maximum cut (3), minimum vertex cover (20), maximum independent set (7), maximum common subgraph (2) and 4-moments portfolio optimization (8) are some fundamental problems, which have been solved specifically by deep Q-learning network.

# 3 Preliminaries

## 3.1 Problem definition

We define the P+O problem as:

$$\min_{X \in C} H^T \cdot X \tag{1}$$

where $X \in \mathbb{R}^n$ are the decision variables, $C \subseteq \mathbb{R}^n$ is the feasible region, $H = (h_1, h_2, \ldots, h_t) \in \mathbb{R}^t$ are the hidden parameters, and $obj(\cdot, \cdot) : \mathbb{R}^n \times \mathbb{R}^t \mapsto \mathbb{R}$ is an objective function, whereby $obj(X, H)$ measures the objective value of decisions $X \in C$ using parameters $H$. Besides, a tuple of feature vectors $A = (\vec{a}_1, \vec{a}_2, \ldots, \vec{a}_t) \in \mathbb{R}^{t \times m}$, related to hidden parameters $H$, is given. Each $\vec{a}_i \in \mathbb{R}^m$ can be used to estimate $h_i$.

## 3.2 Baseline method

We compare our proposed RL approach with the state-of-art P+O methods.

**Smart "Predict, then Optimize" (SPO).**    This paper laid the foundation for P+O, and most of the existing work is based on the new loss function proposed in this paper. Let $s^*(H)$ denote the true optimal solution, $s^*(\hat{H})$ denote the estimated optimal solution. The SPO loss function is:

$$loss_{SPO}\left(\hat{H}, H\right) = \min_{X \in C}\left\{H^T \cdot s^*(\hat{H})\right\} - H^T \cdot s^*(H). \tag{2}$$

Since the set of decision variables and the set of domains are both finite sets, there are only finite solutions of a COP. In other words, SPO loss function is a piecewise constant function and the gradient is identically zero or does not exist. Therefore, a surrogate SPO+ loss function is derived from three upper bounds on SPO loss function:

$$loss_{SPO+}\left(\hat{H}, H\right) = \min_{X \in C}\left\{H^T X\right\}\left(H - 2\hat{H}\right) + 2\hat{H}^T w^*(H) - s^*(H) \tag{3}$$

where $w^*(V)$ denote the optimization oracle. Practically speaking, since $Sol^*(H)$ is usually a singleton, we should expect $w^*(H)$ to be a unique optimal solution.

**SPO Trees (SPOTs).**    This algorithm is proposed for training decision trees using the SPO loss function (14). The objective is to partition the training observations into $L$ leaves, $R_1, \ldots, R_L :=R_{1:L}$, whose predictions collectively minimize the SPO loss function (3). Let $n$ denote the number of the feature-parameter samples, to fit with the tree setting, the purpose of minimize SPO loss function can be written as:

$$\min_{R_{1:L} \in \tau} \frac{1}{n} \sum_{l=1}^{L} \sum_{i \in R_l} (h_i^T s^*(\hat{h}_l) - h_i^T s^*(h_i)). \tag{4}$$

The main difference between SPOT and other decision tree methods is how to split feature. Define $x_{i,j}$ as the $j$-th feature component corresponding to the $i$-th training set observation. Beginning with the entire training set, consider a decision tree split $(j, s)$ represented by a splitting feature component $j$ and split point $s$ which partitions the observations into two leaves:

$$R_1(j, s) = \{i \in \{1, 2, \ldots, n\} \mid x_{i,j} \leq s\} \text{ and } R_2(j, s) = \{i \in \{1, 2, \ldots, n\} \mid x_{i,j} > s\},$$

if variable $j$ is numeric, or

$$R_1(j, s) = \{i \in \{1, 2, \ldots, n\} \mid x_{i,j} = s\} \text{ and } R_2(j, s) = \{i \in \{1, 2, \ldots, n\} \mid x_{i,j} \neq s\},$$

if variable $j$ is categorical. The first split of the decision tree is chosen by computing the pair $(j, s)$ which minimize the following optimization problem:

$$\min_{j,s} \frac{1}{n} \left( \sum_{i \in R_1(j,s)} (h_i^T s^*(\hat{h}_l) - h_i^T s^*(h_i)) - \sum_{i \in R_2(j,s)} (h_i^T s^*(\hat{h}_l) - h_i^T s^*(h_i)) \right).$$

**SPO Forests.**    SPO forests is a methodology for training an ensemble of SPO Trees to boost decision performance (14). SPO Forests are constructed using (greedy) SPO Trees through the same procedure as random forests are constructed using CARTs. Random forests are known to have less variance than individual decision trees, at the price of sacrificing interpretability. To construct an SPO Forest, $B$ SPO Trees are trained on bootstrapped samples of the training dataset, where $B$ represents the number of desired trees in the SPO Forest.

## 4 RL framework

### 4.1 Network architecture

We focus on the traveling salesman problem (TSP) in this paper. Given a directed graph $G = \{V, E\}$ where each edge $e_{ij}$ is associated with a distance $h_{ij}$ and a feature vector $\vec{a}_{ij}$, a source vertex $s$ and a sink vertex $t$. We are concerned with finding a permutation of the edges $\pi$, termed a tour, that has the shortest total distance from source to sink. Suppose that there are $n$ edges in $\pi$, then we define the length of a tour by a permutation $\pi$ as:

$$L(\pi \mid s) = \sum_{k=1}^{n} dist_{\pi(k)} \tag{5}$$

Different from the traditional TSP, distances are hidden here. Our aim is to find a stochastic policy $p(\pi \mid s)$ that can minimize equation (6). Therefore, the parameters of the policy $p(\pi \mid s)$ are expected to been learned according to a given input set of edges $s$ and related features. For example, the given graph may like figure 1(a). The probability of a certain policy can be further factorized as equation (6). In this case, our pointer network can use individual softmax modules to represent the probability of the policy's $k$-$th$ step. The reason for choosing a pointer network is that it allows the model to point to the input sequence's specific position.

$$p(\pi \mid s) = \prod_{k=1}^{n} p(\pi(k) \mid \pi(<k), s) \tag{6}$$

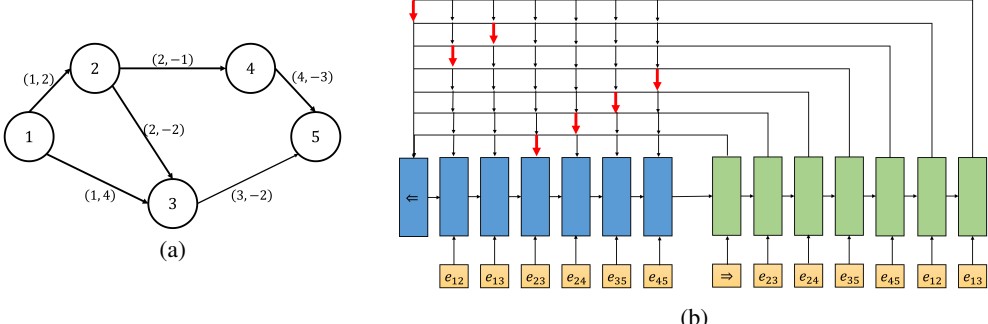

(a)

(b)

Figure 1: (a) is an example for the TSP with unknown distances, the tuple on each edge represents the features related to the distance. (b) is a pointer network architecture introduced by (28).

Motivating by previous work (28; 4; 17; 20; 21; 25), we propose to use pointer network, introduced by (28), to train the model. Our pointer network comprises two recurrent neural network (RNN) modules, encoder and decoder, as shown in 1(b). The input is a sequence of edges features, rather than the cities coordinates. For example, the input sequence of figure 1(a) is $\{(1, 2, 1, 2), (1, 3, 1, 4), (2, 3, 2, -3), (2, 4, 2, -1), (3, 5, 3, -2), (4, 5, 4, -3)\}$. Pointer network has an encoding RNN, which converts the input sequence to a code (blue) that is fed to the generating network (green). At each step, the generating network produces a vector that modulates a content-based attention mechanism over inputs. The output of the attention mechanism is a softmax distribution with dictionary size equal to the length of the input.

### 4.2 Optimization with Actor-critic

We utilize RL as the paradigm for training the pointer network for COPs because of its relatively simple reward mechanisms. Inspired by previous work (4; 25) that apply RL scheme to deterministic COPs, we propose to optimize the parameters of the pointer network using Actor-critic algorithm. Our training objective is the expected tour length which, given an input graph $s$, is defined as

$$J(\theta \mid s) = \mathbb{E}_{\pi \sim p_\theta(.|s)} L(\pi \mid s) \tag{7}$$

The gradient is

$$\nabla_\theta J(\theta \mid s) = \mathbb{E}_{\pi \sim p_\theta(.|s)} \left[ L(\pi \mid s) \nabla_\theta \log p_\theta(\pi \mid s) \right] \tag{8}$$

Then, we subtract a baseline function $b(s)$ that does not depend on $\pi$ and estimates the expected tour length to reduce the variance of the gradients:

$$\nabla_\theta J(\theta \mid s) = \mathbb{E}_{\pi \sim p_\theta(.|s)} \left[ (L(\pi \mid s) - b(s)) \nabla_\theta \log p_\theta(\pi \mid s) \right] \tag{9}$$

Sampling a single tour per graph, i.e., $\pi_i \sim p_\theta(. \mid s_i)$, the gradient in (10) is approximated with Monte Carlo sampling as follows:

$$\nabla_\theta J(\theta \mid s) = \frac{1}{B} \sum_{i=1}^{B} \left[ (L(\pi_i \mid s_i) - b(s_i)) \nabla_\theta \log p_\theta(\pi_i \mid s_i) \right] \tag{10}$$

## 5 Preliminary results

In this section, the preliminary results are presented. In ahead of solving the P+O problem, we first focus on using reinforcement learning and neural networks to solve the COP, more specifically, to solve the conventional TSP problem. The network model in (4) is employed, and benchmark tasks, Euclidean TSP5, TSP10 are considered to be solved.

### 5.1 Setup detail

The training mini-batches of inputs with size 512 are generated, and the model parameters are updated with the Actor-Critic Algorithm. The hidden units for RNN cells are set as 128. The learning rate is 0.001 and is decayed by a factor of 0.96 for every 5000 steps. The network is trained on 2560000 graphs for both TSP5 and TSP10 cases. Points are drawn uniformly at random in the unit square $[0, 1]^2$.

### 5.2 Results

Figure 1 shows the actor loss and critic loss variation for the first 1000 epochs during the training process. The actor loss has a fluctuation initially, while the critic loss drops rapidly at the same time. However, both the actor loss and the critic loss converge to around 0 after hundreds of epochs. This result shows the reinforcement learning and neural networks model has an excellent ability to converge.

The comparisons average tour length difference between the optimal solution and the output path from our model are presented in Figure 2. The corresponding average tour length of the optimal solution and the counterpart from our model are also shown in Figure2. The result includes the first 5000 epochs during the training process for both TSP5 (a) and TSP10 (b) cases. In each epoch, 512 random generated graphs are employed, and the result is the average length/difference of the 512 graphs. As the figure shows, the tour length difference between our model's output and the optimal value decreases fast in the first 1000 epochs. After training around 4000 epochs, our model can achieve about just 4% worse than optimality for TSP10. The same performance with average length difference 0.11 is also validated on testing dataset. Therefore, for both TSP5 and TSP10 cases, our model acquires a comparable performance with the optimality.

## 6 Experiment results

A similar method is applied to P+O TSP Problem. P+O TSP5, TSP20, TSP50, and TSP 100 are investigated in this paper. Instead of basing on the input nodes' coordinates, four features of these two nodes determine the distance between two nodes. Our model results are compared with the count part using random select, linear regression, SPO tree, and SPO forest.

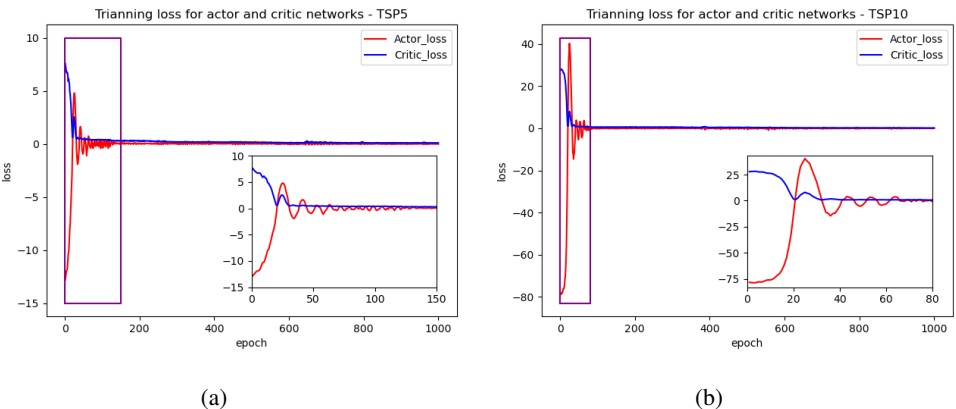

(a)                                                                          (b)

Figure 2: Average actor loss and critic loss vs. training epoch (a) TSP with 5 nodes. (b) TSP with 10 nodes.

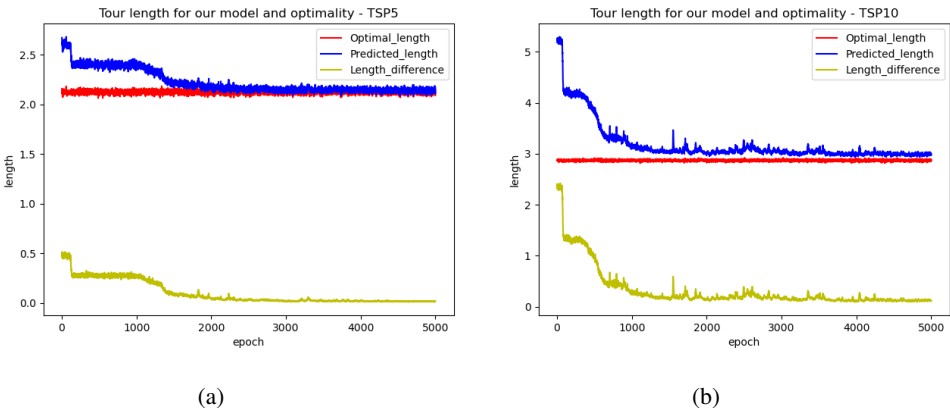

(a)                                                                          (b)

Figure 3: Tour length vs. training epoch (a) TSP with 5 nodes. (b) TSP with 10 nodes.

## 6.1 Setup detail

The employed training method and neural network architecture are the same as described in section 5.1. The distance between two nodes in the graph is based on equation (11).

$$D_{i,j} = 10 \sin \frac{x_i + x_j}{2} \sin \frac{y_i + y_j}{2} + 100 \sin \frac{z_i + z_j}{2} \sin \frac{k_i + k_j}{2} + 110 \tag{11}$$

Where $D_i$ donates the distance between node $i$ and node $j$. $x_i, y_i, z_i$, and $k_i$ are four input features for node $i$.

## 6.2 Results

Figure 4.(a) shows the actor loss and critic loss variation for the first 1000 epochs during the node 5 case training process. Similar to the results in section 5.2, the actor loss has a fluctuation initially, while the critic loss drops rapidly at the same time. However, both the actor loss and the critic loss converge to around 0 after hundreds of epochs.

The predicted average tour length for P+O TSP5 based on our model is presented in Figure 4.(b). We compare our model's first 80000 iteration results with random selection, linear regression, SPO tree, and SPO forest. As the figure shows, random selection and linear regression get the worst performance. SPO tree and SPO forest attain better performance with lengths 530 and 511.38,

respectively. Our RL model achieves the best results with a distance of about 460. Although the optimal value is 422.13, our RL model's output can further approach this value after more training iterations. Therefore, for the P+O TSP5 problem, our RL model beats all other methods.

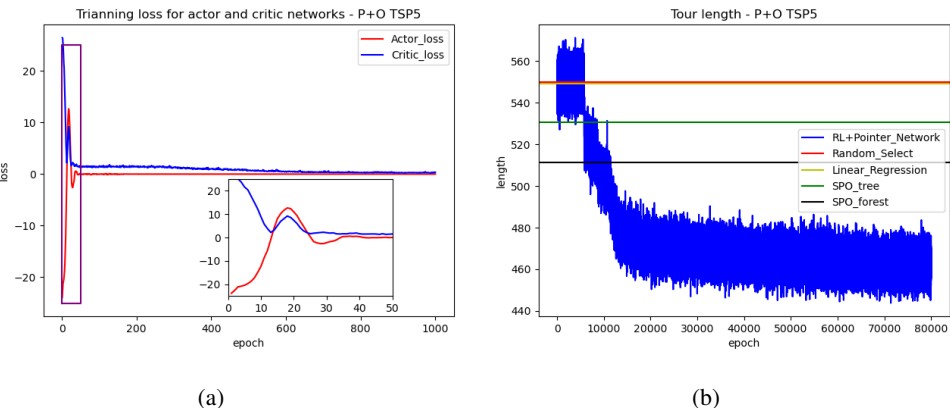

(a)                                                    (b)

Figure 4: Training result for P+O TSP5 (a) Average actor loss and critic loss vs. training epoch. (b) Tour length vs. training epoch.

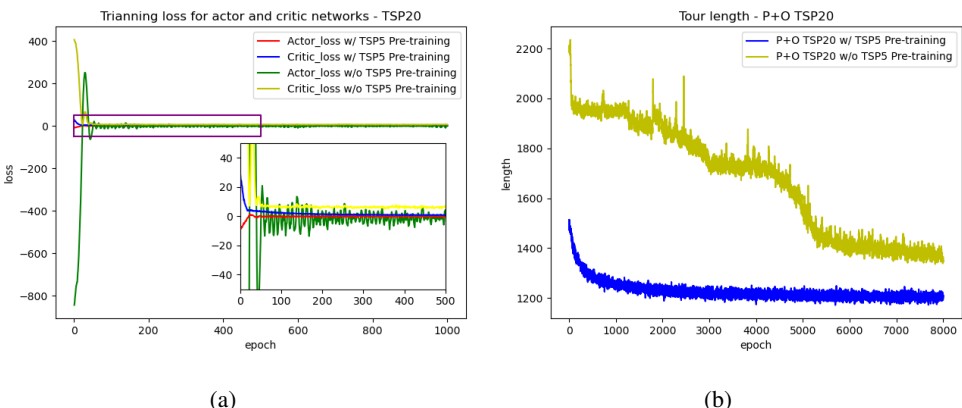

(a)                                                    (b)

Figure 5: Comparison result for P+O TSP20 w/ and w/o pre-training (a) Average actor loss and critic loss vs. training epoch. (b) Tour length vs. training epoch.

Our RL model has another advantage due to the pointer network architecture. The same model can deal with different TSP problem with varying nodes of the input number. This induces two main benefits. On the one hand, we can train our model with simple problem cases while applying the trained model on a more challenging task. For instance, table 1 illustrates the results that the model is trained with TSP5 and used on TSP20, 50, and 100. On the other hand, we can utilize the model trained with a simple case to speed up the more formidable task's training process. Figure 5 gives an example of comparing the training process of TSP20 with/without the pre-training of TSP5.

As Table 1 shows, although the RL model is trained with the only TSP with five nodes, it performs better than SPO Tree and SPO Forest when applied on TSP20, 50, and 100. This reveals the high flexibility and reliability of our RL model. Moreover, figure 5. (a) shows both the actor loss and critic loss converge faster for the case using TSP5 pretraining. Figure 5.(b) display that without TSP5 pre-training, the output of our RL model for P+O TSP20 is similar to the random selection. Simultaneously, it also needs 5000 iterations to get similar performance as the first several iterations from the model with TSP5 pretraining. This also shows that the training process of a more challenging task can speed up using the pre-trained model of simpler tasks.

Table 1: Tour length Results for Different Methods

| Testing Problem | RL trained with TSP5 | Random Selection | SPO Tree | SPO Forest | Optimal |
|---|---|---|---|---|---|
| P+O TSP5 | 455.95 | 550 | 530 | 511.38 | 422.13 |
| P+O TSP20 | 1480.36 | 2200 | 2059.27 | 1919.63 | 764.57 |
| P+O TSP50 | 2475 | 5500 | 2955.113 | 2726.5 | 1111.98 |
| P+O TSP100 | 5929.88 | 11000 | 7943.4 | 7387.5 | 1633.78 |

## 7   Conclusion

This paper proposed a new approach to solving the P+O problems, which combine the pointer network and reinforcement learning. To verify the proposed approach, we employ it to tackle the P+O TSP. Baseline methods, including Random Selection, SPO Trees, and SPO Forest, are also used to evaluate the final result. Numerical result shows that the proposed RL model achieves the best performance among all the competitors. The proposed approach can even beat other methods for solving P+O TSP20, TSP50, and TSP100 Problem using the model only trained with TSP5. RL model can deal with TSP problems with different input nodes benefiting from the pointer network's special architecture. This renders two main advantages of our RL model. On the one hand, we can train our model with a simple problem case while applying the trained model on a more challenging task; on the other hand, the training process for a more challenging problem can speed up by utilizing the model trained with a simper task.

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
