# OpenReview forum: "Reinforcement Learning for Predict+Optimize"
_CUHK.edu.hk/2021/Course/IERG5350_

### Official Review · AnonReviewer2 · 2020-12-17
**Nice work...**

**Rating:** 9
**Confidence:** 3

**Review:**

Summary:
* This paper proposes an RL approach to tackle the Predict+Optimize (P+O) problem, in particular, the traveling salesman problem (TSP), and compare it with several state-of-the-art P+O methods, Smart “Predict, then Optimize” (SPO), SPO Trees (SPOTs), and SPO Forests. Specifically, the proposed method leverages the actor-critic paradigm to train the pointer network for solving TSP. The experiment results show the excellent performance of the proposed approach over all other baselines. Surprisingly, the trained model exhibits impressive generalizability in that the model trained with a simple problem can be applied to a harder one while achieving comparable performance (i.e., train on TSP5 and test on TSP 20, 50, 100). In addition, the pre-trained model on a simple problem (TSP5) can speed up and even improve the convergence of model training on a harder one (TSP20).

---

### Official Review · AnonReviewer1 · 2020-12-18
**A good thought of tacking Predict+Optimize problem**

**Rating:** 8
**Confidence:** 4

**Review:**

Summary:
In this paper, the authors propose to use reinforcement learning with pointer network  for tacklling the traveling salesman problem.
Moreover, their proposed  pointer network architecture can deal with different TSP problem with varying nodes of the input number. The result demonstrate that their RL achieves the best results compared to traditional baseline methods with a distance of about 460.

Pros:
Their paper is well organised and written. The main contribution is to bring the neural network and RL into the optimazation problem.

Cons:
This paper use Actor-critic method to train the network. Less work about RL algorithm: the authors did not explain clearly why they use this method, and I can not see the comparison with other algorithm (such as PPO, TD3)

Which need to be mentioned is that, the video is a little longer (8min) than required (5min), it would be better to condense the talk.

---

### Official Review · AnonReviewer3 · 2020-12-19
**A RL frame work to solve TSP**

**Rating:** 7
**Confidence:** 3

**Review:**

This paper focus on solving the travel salesman problem(TSP) by leveraging the RL technique. To tackle the non-differentiable problem of supervised SPO, inspired by deterministic COP reinforcement technique author’s proposed a RL frame work to solve the P+O TSP. Instead of leaning the coordinate and then minimize the objective. This method directly leaning a policy from the edge feature.  The experimental results shows the model outperform the other methods.

I think the novelty is enough. The related works is well-organized. The video presentation is clear and provide a panoramic view of this paper.

### Some questions:

1. I found that the feature can easily calculate the distance between two points. Why model used the feature instead of calculated distance?

2. The last experiment attract me a lot, why the pertained model can improve the model by a large marginal. Is the model without pertained stuck in the how to learn a high level information(e.g, distance)?


### Some suggestions:

1. The introduction part can be longer. Current half page is not cover the novelty of this paper and is not informative.

2. More detail of pointer network is needed. I think the figure 1 (b) need to show the detail of How does the decoder RNN modulate the encoder RNN. And Why the encoder need to be modulated by the decoder?

3. There are too many elements in the figure 1(b) are not identified. e.g., What the meaning of red arrow above RNN. What dose the arrow in leftmost encoder RNN stand for?

4. More insight or the reason of why the RL is better in this area can be given. (e.g., why the RL is better than the original SPO)

5. Maybe the graph neural network (GNN) is more suitable for this problem. I didn’t get the point of using the recurrent neural network. I think RNN is more suitable for the temporal or sequential input. In this case is just a set of unordered points.

6. Giving more detail of actor and value network.  I didn’t find text that describe the actor and value network’s specific structure and the relation with the pointer network.